# Contribution Ratio Assessment of Process Parameters on Robotic Milling Performance

**DOI:** 10.3390/ma15103566

**Published:** 2022-05-16

**Authors:** Jing Ni, Rulan Dai, Xiaopeng Yue, Junqiang Zheng, Kai Feng

**Affiliations:** 1School of Mechanical Engineering, Hangzhou Dianzi University, Hangzhou 310005, China; nj2000@hdu.edu.cn (J.N.); dairulan@hdu.edu.cn (R.D.); 42360@hdu.edu.cn (J.Z.); fengkai319@foxmail.com (K.F.); 2School of Mechanical Engineering, Hangzhou Dianzi University Information Engineering College, Hangzhou 311305, China

**Keywords:** contribution ratio, process parameters, milling load, surface quality, vibration

## Abstract

Robotic milling has broad application prospects in many processing fields. However, the milling performance of a robot in a certain posture, such as in face milling or grooving tasks, is extremely sensitive to process parameters due to the influence of the serial structure of the robot system. Improper process parameters are prone to produce machining defects such as low surface quality. These deficiencies substantially decrease the further application development of robotic milling. Therefore, this paper selected a certain posture and carried out the robotic flat-end milling experiments on a 7075-T651 high-strength aeronautical aluminum alloy under dry conditions. Milling load, surface quality and vibration were selected to assess the influence of process parameters like milling depth, spindle speed and feed rate on the milling performance. Most notably, the contribution ratio based on the analysis of variance (ANOVA) was introduced to statistically investigate the relation between parameters and milling performance. The obtained results show that milling depth is highly significant in milling load, which had a contribution ratio of 69.25%. Milling depth is also highly significant in vibration, which had a contribution ratio of 51.41% in the X direction, 41.42% in the Y direction and 75.97% in the Z direction. Moreover, the spindle speed is highly significant in surface roughness, which had a contribution ratio of 48.02%. This present study aims to quantitatively evaluate the influence of key process parameters on robotic milling performance, which helps to select reasonable milling parameters and improve the milling performance of the robot system. It is beneficial to give full play to the advantages of robots and present more possibilities of robot applications in machining and manufacturing.

## 1. Introduction

The application of industrial robots for milling processing has the advantages of large processing range, flexibility and speed compared with traditional multi-axis milling machining centers [1,2], and it occupies an important position in the fields of aerospace, energy and transportation, for example, the machining of aircraft wings, aircraft fuselages and other aviation structural parts in the aerospace field. The robot can perform milling tasks such as chamfering, deburring and polishing by clamping milling cutters, rotary burrs and other tools [3,4,5]. However, industrial robots have low system rigidity due to their own open-chain multi-rod serial structure. Improper milling process parameters will easily cause large-scale fluctuations in machining accuracy, and ultimately affect the overall performance of the product, including wear resistance and fatigue resistance and further processing and assembly [6]. At the same time, when the robot is applied in the cutting of difficult-to-cut materials such as titanium alloy [7], the robot milling system is prone to chattering, which leads to a decrease in processing accuracy, and even serious consequences such as product scrap and robot damage [8]. Therefore, choosing reasonable milling process parameters can greatly reduce the vibration of the robot and improve the milling performance of the robot system, which helps to overcome the deficiencies of industrial robot milling [9].

In the past ten years, robotic drilling technology has developed more maturely in the field of robotic processing. However, the research on robotic milling technology has mainly focused on improving the robotic machining accuracy and the vibration suppression during the milling process, and the research and comprehensive analysis on the influence of milling parameters on milling load, surface quality and vibration during robotic milling are insufficient. Therefore, it is necessary to study the effects of process parameters on robotic milling performance [10,11,12,13].

In the research on the industrial robot milling processing system, several studies have been performed. Kihlman et al. developed a robotic spiral milling system for difficult-to-process materials such as titanium alloys and composite materials. The results showed that spiral milling can reduce the axial force in the cutting process and improve the machining quality [14]. Hao et al. conducted an experimental study of stability prediction for high-speed robotic milling of aluminum. Modal tests were conducted on a milling robot by using a laser tracker and a displacement sensor [15]. Yin et al. studied the machining error prediction and compensation technology for a stone-carving robotic manipulator. The feasibility and effectiveness of the proposed compensation technology were verified by an experiment using the KUKA-240-2900 SCRM system [16]. Chen et al. introduced a milling force model for robotic milling of cortical bone, and analyzed the influence of bone material anisotropy on the milling force. At the same time, they also considered the low stiffness of the robot and introduced it into the milling force model [17]. Tunc et al. conducted an experimental study on the dynamics of a hexapod robot for mobile machining. The results can be generalized to mobile machining with hexapod robots [18].

In recent years, several studies have been conducted on the influence of process parameters on milling performance. As for the milling experiment investigations conducted in traditional CNC machining centers, Abou-EI-Hossein et al. established statistical laws between different milling parameters and milling forces based on a large amount of experimental data [19]. Kull Neto Henrique et al. conducted an experimental study on the milling component force and surface roughness when milling molds for different milling strategies and tool overhangs. The results of variance analysis showed that the mold surface roughness is directly related to the radial force [20]. Lu et al. developed a surface roughness model considering tool vibration, which can be used to estimate surface roughness under different machining parameters and tool geometry parameters during the micro-milling process [21]. Ratnam et al. investigated the influence of process parameters on machining performance during turn-milling processes by using the Taguchi method. Experimental results show that feed rate, tool speed and cutting depth are of different importance in generating surface roughness and surface hardness [22]. Salur et al. studied the influence of MQL and dry conditions on the end milling process, and found that the cutting environment had the greatest influence on power consumption, while cutting speed had an important influence on tool wear [23].

In the robotic milling research, ABB found that when the robot is in a certain posture and the cutting depth reaches a certain value, milling chatter with a larger amplitude will occur, which will damage the machined parts [24]. Zhang et al. analyzed the influence of different spindle speeds and machining methods on the vibration during the milling process when a new milling tool was used for robot milling, and optimized the spindle speed, feed speed and machining method by the response surface method [25]. Lei et al. proposed a new method to reduce the mode coupling chatter of the robot. They actively suppressed the chatter generated during the milling process of the robot by installing a special vibration suppressor on the spindle [26]. Jayakrishnan et al. obtained the optimal machining parameters for robotic end milling by using the Taguchi–Grey relational method. They found that the tool path during robotic machining has the greatest influence on machining performance [27]. Chen et al. studied the posture and feed orientation optimization in the robotic milling process based on the stiffness performance index. The normal stiffness performance index (NSPI) of the surface, which is derived from the comprehensive stiffness performance index (CSPI), was proposed to evaluate the robot stiffness performance for a given posture [28]. Tunc et al. proposed a new method to study the effects of tool path and feed direction on chatter in order to improve material removal rate without chatter during robotic milling. Simulation and experimental results have confirmed the effectiveness of the proposed method [29]. Sun et al. investigated the chatter stability of robotic rotary ultrasonic milling and developed an analytical model of stability. The analysis results indicate that stability region of RRUM is improved by 133% compared with robotic conventional milling (RCM) [30]. He et al. studied the optimization algorithm of the milling path through a new stiffness orientation method, which is of great significance for optimizing the milling path of the robot, reducing machining chatter and thus improving the machining stability of robotic milling [31]. By adopting an optimal control method related to robotic posture, Nguyen et al. were able to actively suppress tool tip vibrations during the robotic milling mainly caused by the characteristics of milling forces. This method was proven to be effective in improving robotic machining accuracy [32]. Li et al. discussed the influence of tool path and workpiece position on machining chatter during planar milling. The results show that the surface quality obtained in different directions is significantly different due to the different machining stiffness of robots in different directions [33]. In order to improve the quasi-static performance of robotic milling, Qin et al. obviously reduced the tool offset during robot machining by optimizing the workpiece pose. At the same time, the optimization of workpiece poses can also reduce the change of the tool offset along the processing path [34].

According to the abovementioned literature items, the majority of researchers have studied the effects of material properties, milling methods and the robot itself on the robotic milling performance. There are few studies of percentage contribution on the influence of milling parameters such as milling depth, spindle speed and feed rate on the milling performance of the robot in a certain posture. Therefore, the comparative experiments of flat-end milling processing of a 7075-T651 high-strength aeronautical aluminum alloy were carried out on the robot under dry conditions in a common posture. Milling depth, spindle speed and feed rate were chosen as process parameters. Next, the effect of process parameters on robotic milling performance, such as milling load, surface quality and vibration, was evaluated by ANOVA. Moreover, the milling load empirical model of the relationship between the milling load and process parameters was established. Therefore, the main purpose of this article is to investigate the influence of the milling parameters on the robotic milling performance using the contribution ratio. The research results can provide some guidance for the selection of process parameters when the six-degrees-of-freedom industrial robot is milling aluminum alloy materials, which is conducive to improving the processing quality and the stability of the robotic milling and increasing the application possibilities of the robot in more fields.

## 2. Experiment

### 2.1. Experimental Setup

The experimental tests were carried out on a 7075-T651 aluminum alloy using a cemented carbide three-blade milling cutter under dry conditions. The total length of the tool was 75 mm, the blade length was 20 mm with a diameter of 6 mm and the helix angle was 55°. Figure 1 shows the schematic diagram of the experimental setup. The milling tool rotates clockwise and feeds in the negative direction of Y. The dimensions of the workpiece were 100 mm × 45 mm × 15 mm. The nominal chemical composition and material properties of 7075-T651 are listed in Table 1 and Table 2, respectively. The Mitsubishi RV-4FRL-1D-S11 six-axis industrial robot and NAKANISHI spindle were used for milling experiments. The specifications of the robot and spindle are shown in Table 3. The three milling load components (radial load $F_{x}$, main milling load $F_{y}$ and axial load $F_{z}$) were acquired with a three-dimensional piezoelectric sensor (PZT. sensor; type: ME-K3D120). The vibration accelerations were measured with a three-dimensional acceleration sensor (ACC. sensor; type: SD14N20) installed on the spindle, and an UTEKL test system was used for acquiring the milling load signal and vibration signal of the robot during milling with its acquisition frequency of 1280 Hz. Mitutoyo SJ-210 measuring instrument was employed to measure and analyze the roughness of the processed surface. In order to observe the morphology of the processed surface, a KEYENCE VW-9000 high-speed optical microscope was also used.

Before investigating the effect of process parameters, exploratory experiments were first conducted on the influence of the robot structure. Among them, the milling depth, spindle speed and feed rate remained the same, and their values were 0.15 mm, 3500 r/min and 45 mm/min, respectively. There were 12 experiments in total, and three repetitive processing experiments were carried out along the positive and negative directions of X and Y, respectively. Figure 2 shows the machined surface results for milling by the robot in different directions.

According to the analysis of the machined surface, there is a noticeable difference in the machined surface quality when the robot feeds in different directions. On the whole, the machined surface quality feeding in the Y direction is better than feeding in the X direction. The vibration in the Y direction is not as pronounced as the vibration in the X direction. Moreover, when the robot feeds in the negative direction of X, the machined surface quality is better than feeding in the positive direction of X. The same is true for the Y direction. It is known from the literature that the stiffness in different directions in the robot WCS is different. This is because the feed direction of the robot during milling largely affects the value of the diagonal elements of the stiffness matrix of the robot end. Furthermore, the value of the robot end stiffness matrix is related to the change of the robot Jacobian matrix. The Jacobian matrix changes with the robot posture, so the stiffness performance of the robot end is affected by the robot pose. This explains the obvious vibration marks on the machined surface when the robot feeds in the positive direction of X. In summary, the negative direction of Y with better processing performance was selected for further research in this paper.

This research considered the three influencing factors of milling depth, spindle speed and feed rate. According to literature data [25], as well as to the robot’s limitations, the selected levels of each influencing factor are shown in Table 4. Among them, spindle speed n is a technological parameter needed for the setting of spindles. The normal working speed of the spindle is about 80% of its maximum speed. In this study, the rated speed of the spindle is 7500 rpm and its 80% value is 6000 rpm. When the spindle speed is about 5500 rpm, the working noise is quite obvious. Therefore, the level of e spindle speed was set as 2500, 3000, 3500, 4000 and 4500 rpm. The actual value of cutting speed, $v_{c}$, corresponding to spindle speed is 47.1, 56.5, 66.0, 75.4 and 84.8 m/min ($v_{c}=\pi nd/1000$; d is the tool diameter). In this paper, a statistical design of experiments based on Taguchi’s Orthogonal Array (OA) was adopted for experimentation, and the $L_{25}\left(5^{6}\right)$ orthogonal experiment was designed.

### 2.2. Evaluation Method

In order to compare the degree of influence of each process parameter on the milling performance, number pairs of process parameters, $p\left(i,j,k\right)$, was used for the analysis, expressed as:(1)$$p\left(i,j,k\right)=\left(v_{f}\left(i\right),\;\;n\left(j\right),\;\;a_{p}\left(k\right)\right)$$
(2)$$p=\left\{p\left(i,j,k\right)\right\}$$
(3)$$v_{f}={\left\{\left.v_{f}\right|v_{f}=v_{f}\left(i\right)\right\}}_{i=1,2,3,4,5}=\left\{15,\;30,\;45,\;60,\;75\right\}$$
(4)$$n={\left\{\left.n\right|n=n\left(j\right)\right\}}_{j=1,2,3,4,5}=\left\{2500,\;3000,\;3500,\;4000,\;4500\right\}$$
(5)$$a_{p}={\left\{\left.a_{p}\right|a_{p}=a_{p}\left(k\right)\right\}}_{k=1,2,3,4,5}=\left\{0.05,\;0.10,\;0.15,\;0.20,\;0.25\right\}$$
where $p\left(i,j,k\right)$ represents the process parameter when the feed rate is $v_{f}\left(i\right)$, the spindle speed is $n\left(j\right)$ and the milling depth is $a_{p}\left(k\right)$. In this study, i,  j   and k represent the different levels of each process parameter.

Generally, the ANOVA can be used to quantitatively estimate the influence of each factor on the response results [35,36]. The contribution ratio of each factor can be obtained by separating the total variation of the response results. Therefore, in order to obtain the proportion of the influence of the level change of each process parameter on the fluctuation of the response data during the robotic milling experiments, the contribution ratio was introduced for assessment. The factors with the largest contribution ratio are significant factors, and those similar to the error contribution ratio are considered insignificant. The contribution ratio is expressed by Formula (6). F test is a statistical test method proposed by R.A. Fisher, which is mainly applied in ANOVA [19,22].
(6)$$\mathrm{Contribution}\;\mathrm{ratio}\left(\%\right)=\frac{{\mathrm{MS}}_{B}}{{\mathrm{MS}}_{T}}$$
where ‘${\mathrm{MS}}_{B}$’ is the average sum of square between groups and ‘${\mathrm{MS}}_{T}$’ is the average sum of the square in total.

## 3. Results and Discussion

### 3.1. Milling Load

In the milling process, milling load is a critical physical parameter, which directly affects robotic milling performance. Therefore, it is vital to study the contribution ratio of process parameters on milling load. A series of repetitive robotic milling experiments have been performed on 7075-T651 aluminum alloy based on the orthogonal experiment table designed in Table 4. The complete obtained results for milling load, surface roughness and vibration are given in Table 5. As shown on the right side of Figure 1, for each experiment, the signal of the three milling load components were recorded with a piezoelectric sensor. Then the three milling load component signals recorded by the piezoelectric sensor were amplified by the amplifier and then transmitted to the UTEKL data acquisition instrument. The sampling frequency of the UTEKL system was set as 1280 Hz and then the acquired signals were analyzed and processed by the signal analysis software in the computer. Eventually the dynamic response diagram of the milling load over time was obtained. All experimental results were in good consistency. Therefore, a set of relatively stable dynamic curves was randomly selected to analyze the milling load. The stable stage in the response diagram was selected for filtering and noise reduction processing to obtain the average value of milling load. As shown in Figure 3, the stable milling stage from 50 to 70 s in Experiment 1 was selected for calculation. When the feed rate was set to other levels in Table 4, the values of $t_{1}$ and $t_{2}$ were selected according to the table on the right side of Figure 3. Finally, the changes in the milling load under different process parameters were evaluated by the resultant load $F_{R}$.

The contribution ratio based on the analysis of variance (ANOVA) was introduced to statistically investigate the relation between parameters and milling load. Table 6 shows the ANOVA analysis results for milling load. Among them, E denotes the error; SS denotes the sum of squares of each factor; MS denotes the average sum of squares; D denotes the degree of freedom; $F_{0.05}\left(4,12\right)=3.26$. F is a statistic. The results of variance analysis showed that the order of the significant influence on milling load is: milling depth—1; feed rate—2; spindle speed—3. Milling depth $a_{p}$ is a highly significant influencing factor (HS), while feed speed $v_{f}$ is a significant influencing factor (S), and spindle speed n is not significant (NS). The contribution ratio of the above process parameters on the milling load is 69.25%, 5.78% and 23.32%, respectively. The milling depth has the maximum contribution ratio on the milling load and its contribution ratio is much larger than the spindle speed and feed rate. The contribution ratio of the spindle speed is similar to the error contribution ratio and it is considered insignificant. The main reason for the largest contribution ratio of milling depth is that the variation of milling depth directly changes the unit cutting area, while the variation of feed rate and spindle speed is to change the material removal thickness per unit time by changing the milling thickness per tooth, thereby indirectly changing the unit cutting area [37].

Based on the literature [37], this paper establishes a general form between milling load and milling parameters:(7)$$F_{R}=Ca_{p}{}^{b_{1}}n^{b_{2}}v_{f}{}^{b_{3}}$$
where $F_{R}$ denotes the cutting force, N; $a_{p}$ denotes the axial cutting depth, mm; n denotes the spindle speed, $\left(r\cdot \min^{-1}\right)$; $v_{f}$ denotes the feed rate, $\left(\mathrm{mm}\cdot \min^{-1}\right)$; C, $b_{1}$, $b_{2}$ and $b_{3}$ are constants. Based on the principle of the least square method, according to the milling load data in Table 5, MATLAB was applied to perform the calculation. The result of C=85.693, $b_{1}=0.779$, $b_{2}=-0.404$, $b_{3}=0.384$ was acquired. According to the model, the milling load under different milling parameter combinations was calculated, and the calculated values were compared with the experimental values in Table 5, as shown in Figure 4. It can be seen from Figure 4 that the prediction model is highly significant and fits well with the actual situation. The analysis of variance was used for testing to judge the degree of fitting of the model [38]. It is known that $F=385.83,\;R^{2}=0.92$. Under the given significance level α=0.05, $F_{0.05}\left(3,21\right)=3.07$, $F\gg F_{0.05}\left(3,21\right)$, so the regression model is credible. Therefore, using this model to predict the milling load when milling the 7075-T651 aluminum alloy has a high degree of credibility.

According to the analysis of the milling load model, we can intuitively see the degree of influence of each milling parameter on the milling load. The milling load shows an increasing trend with the increase of the milling depth $a_{p}$ and the feed speed $v_{f}$, and the influence of $a_{p}$ is more obvious. However, as the spindle speed n increases, the milling load decreases and the spindle speed has the least influence on it. This is consistent with the primary and secondary relationship of the influence of each process parameter in Table 6 on the milling load. To sum up, the milling depth is most significant in the milling load compared to spindle speed and feed rate.

### 3.2. Surface Quality

The surface quality of the machined workpiece directly reflects the milling performance of the robot. Surface morphology and surface roughness, as two major parameters reflecting surface quality, can evaluate the surface quality of the workpiece qualitatively and quantitatively, respectively [39]. In order to study the degree of influence of each process parameter on the surface morphology, the workpieces milled by the robot in Section 3.1 were observed under a high-speed optical microscope and acquired the processed surface morphology under each set of milling parameters. For each observation, the middle part of the processing area was selected as the sampling area (Figure 5a); the summary of surface morphology in the sampling area is shown in Figure 5b. Among them, the brightness of the images is to highlight the details of the processed surface, and has nothing to do with the processed material itself. All experiments use the same batch of materials. Further, in order to intuitively analyze the contribution ratio of each process parameter on the surface roughness, the arithmetic mean roughness values of the above-mentioned processed surface were measured by a roughness-measuring instrument with a precision of 0.05 mm. Among them, the cutoff length and sampling length were taken as 0.8 mm and 5 mm, respectively. In order to reduce the measurement error, five points were evenly selected along the feed direction on the measured surface for measurement, and finally the average value of the five results was taken as the experimental value. The obtained results for surface roughness are given in Table 5.

Figure 5b shows the processed surface morphology under 25 sets of milling parameters, which were arranged in the same way as the Taguchi orthogonal experiment. For $p\left(5,1,2\right)$, $p\left(4,1,3\right)$, $p\left(3,1,4\right)$, $p\left(5,2,4\right)$, $p\left(2,3,4\right)$, $p\left(2,1,5\right)$, $p\left(4,2,5\right)$, etc., the tool marks are very serious. The processed surface morphology is poor and there are obvious top-burrs on both sides of the milling groove [40]. The lack of support when cutting the top edges of the material is the main cause of the top-burr formation. As shown in Figure 5b, the number of top-burrs on the workpiece varies greatly under different milling parameters.

As shown in Figure 5b, the surface morphology for level i, $p\left(1,j,k\right)$, $p\left(2,j,k\right)$, $p\left(3,j,k\right)$, $p\left(4,j,k\right)$ and $p\left(5,j,k\right)$, reveals a trend of slightly worsening with the increase of the feed rate; the changes of tool marks and top-burrs are not clear. The surface morphology for level j, $p\left(i,1,k\right)$, $p\left(i,2,k\right)$, $p\left(i,3,k\right)$, $p\left(i,4,k\right)$ and $p\left(i,5,k\right)$, shows a great trend of improvement with the increase of the spindle speed. Comparing the five processed surface images corresponding to $p\left(i,1,k\right)$ and $p\left(i,5,k\right)$, respectively, the changes of tool marks and top-burrs are extremely obvious. Finally, the surface morphology for level k, $p\left(i,j,1\right)$, $p\left(i,j,2\right)$, $p\left(i,j,3\right)$, $p\left(i,j,4\right)$ and $p\left(i,j,5\right)$, shows a tendency to deteriorate with the increase of the milling depth. Comparing the five surface images corresponding to $p\left(i,j,1\right)$ and $p\left(i,j,5\right),$ respectively, the changes of tool marks and top-burrs are obvious. In summary, the spindle speed seems to have the greatest influence on the workpiece surface morphology, followed by the milling depth, and the feed rate has the least effect.

Take Experiments 3, 8, 13, 18 and 23 as examples; the processed surface morphology is shown in Figure 6. The middle of the picture shows the machined surface after 200 times magnification, and the bottom side shows the 3D morphology of the machined surface after 200 times magnification. The surface after milling has obvious tool marks. The evenly distributed tool marks reflect the movement trajectory of the cutting edge of the milling cutter. At the same time, it can be seen that the surface quality at the beginning of milling is poor, and then gradually stabilizes until the tool is completely removed.

According to the experimental results of surface roughness in Table 5, it is first visually analyzed and processed to obtain the degree of influence of each milling parameter on the surface roughness. The results are given in Table 7. Among them, $k_{i}$ is the mean value of the experimental values of each factor at different levels (i = 1,2,3,4,5). The surface roughness increases as the milling depth $a_{p}$ and the feed speed $v_{f}$ increases. The influence of milling depth $a_{p}$ on the surface roughness is more visible compared with feed speed $v_{f}$. The range of variation is 0.546 μm to 1.842 μm and 0.817 μm to 1.621 μm, respectively. The influence of spindle speed n on the surface roughness is sharp. With the increase of n, the surface roughness value decreases greatly. The range of variation is between 0.601 μm and 2.282 μm. The reason for such surface roughness reduction is that the increase of the spindle speed brings higher milling speed, which will reduce the plastic deformation during the milling process, and the built-up edge and burrs will also be reduced [25,41].

To further investigate the effect of process parameters on the surface roughness, the values of feed per tooth $f_{t}$ under each experiment were calculated by Formula (8).
(8)$$f_{t}=\frac{v_{f}}{n\cdot N}$$
where N is the number of teeth. The number of teeth used in this test was 3.

The feed per tooth represents the actual working conditions of the cutting edge. The results of the calculation are shown in Table 8. The effect of feed per tooth on surface roughness is demonstrated in Figure 7. As shown in Figure 7a–e, it is known and obvious that the value of surface roughness decreases with increasing cutting speed for the same value of feed per tooth (see in Figure 7b,e). However, surface roughness also depends on feed per tooth; it mainly increases with feed per tooth for the stability process of cutting (see Figure 7a,e). Figure 7 shows the combined effect of feed per tooth and cutting speed on surface roughness.

Similarly, the contribution ratio based on the ANOVA was introduced to statistically investigate the relation between parameters and surface roughness. Table 9 shows the ANOVA analysis results for surface roughness. The results of variance analysis showed that the order of the significant influence on the surface quality is: spindle speed—1; milling depth—2; feed rate—3. Spindle speed n and milling depth $a_{p}$ are highly significant influencing factors (HS), and feed speed $v_{f}$ is a significant influence factor (S). The contribution ratio of the above process parameters on the surface roughness is 35.83%, 48.02% and 13.18%, respectively. The spindle speed has the maximum contribution ratio on the surface roughness and its contribution ratio is slightly larger than the milling depth. The contribution ratio of the feed rate on the surface roughness is very small compared with the former two. Similar results are reported by studies on the effect of process parameters on surface roughness [42]. The spindle speed (cutting speed) has found to be the most effective parameter on surface roughness. It is known that an increase in cutting speed will increase the cutting temperature and reduce the coefficient of friction between the tool and the chip, thereby reducing the surface roughness. At the same time, the increased temperature can also reduce the adverse effects of a built-up edge (decreasing the cutting thickness can also reduce the occurrence of a built-up edge). Additionally, the effect of spindle speed on the surface roughness is related to the formation of chips. The chip formation rate is slow at low speeds and gets faster at high speeds. This results in the chips being in contact with the newly formed surface for a shorter time, and there is small tendency for chips to wrap back to the new surface compared to the low speed. The chip formation process is also influenced by the shear length in the shear zone. The shear length is related to the undeformed chip thickness and the shear angle [43]. The shear angle is large at high cutting speed. This leads to a smaller shear length. Therefore, the chip will break away with less plastic deformation, which in turn preserves the machined surface properties [44].

To sum up, the spindle speed is most significant in the surface quality compared to milling depth and feed rate. In actual machining, a higher spindle speed should be selected under the condition of tool life-permitting for obtaining better machined surface quality.

### 3.3. Vibration

Excessive vibrations during robotic milling have an important impact on milling stability. Thus, it is very necessary to study the influence of process parameters on robot vibration. The obtained results for vibration acceleration are shown in Table 5, and each result was obtained by calculating the average of maximum values in the vibration acceleration curves. The intuitive analysis results of the influence of each milling parameter on the vibration acceleration are shown in Table 10.

It can be seen from Table 9 that when the robot mills in the horizontal direction (Y direction), the vibration in the Z direction is the largest. Vibration acceleration $a_{x}$,$a_{y}$ and $a_{z}$ show an increasing trend with the increase of milling depth $a_{p}$ and feed speed $v_{f}$, and show a decreasing trend with an increase of spindle speed n. The influence of $a_{p}$ and n is more obvious. For vibration acceleration $a_{x}$, the range of variation is 2.764 g to 13.907 g and 3.380 g to 13.236 g, respectively. For vibration acceleration $a_{y}$, the range of variation is 2.395 g to 13.226 g and 3.515 g to 14.346 g, respectively. For vibration acceleration $a_{z}$, the range of variation is 4.479 g to 38.338 g and 13.613 g to 30.420 g, respectively. However, as the feed speed $v_{f}$ increases, the overall variation of vibration acceleration $a_{x}$, $a_{y}$ and $a_{z}$ is not obvious. The range of variation is 5.343 g to 9.991 g, 5.418 g to 10.032 g and 14.412 g to 26.156 g, respectively.

Table 11 shows the ANOVA analysis results for vibration acceleration. The results of variance analysis showed that the order of the significant influence on the vibration acceleration is: milling depth—1; spindle speed—2; feed rate—3. As for the vibration acceleration generated in the X direction $a_{x}$, milling depth $a_{p}$ and spindle speed n are highly significant influencing factors (HS), and feed speed $v_{f}$ is not significant (NS). Similarly, as for the vibration acceleration generated in the Y direction $a_{y}$, milling depth $a_{p}$ and spindle speed n are also highly significant influencing factors (HS), and feed speed $v_{f}$ is also not significant (NS). The difference is that for the vibration acceleration generated in the Z direction $a_{z}$, milling depth $a_{p}$ is a highly significant influencing factor (HS), while spindle speed n is a significant influencing factor (S) and feed speed $v_{f}$ is not significant (NS). The contribution ratio of the above process parameters on vibration acceleration in the X direction and the Y direction is basically equivalent, being 51.41%, 36.55% and 8.04% and 47.17%, 41.42% and 8.06%, while the contribution ratio for vibration acceleration in the Z direction is 75.97%, 13.43% and 8.16%, respectively.

From the above analysis, it is known that the significance and contribution ratio of the above process parameters on vibration acceleration is different in different directions. In detail, the significance and contribution ratio of the process parameters on vibration acceleration in the X direction and the Y direction is basically equivalent, but there is a big difference compared with the Z direction. The significance of milling depth $a_{p}$ on axial vibration is quite different from the vibration in the feed direction and the vertical feed direction, and its contribution ratio is much larger than the other two directions. Axial vibrations are usually neglected in traditional CNC machines. However, they may significantly affect the stability of the process in industrial robots. According to Mohammadi et al. [45], the numerical case studies show that axial vibrations may cause the feed-generated and edge forces to affect the stability of regenerative vibrations in robotic milling.

In summary, as shown in Table 6, Table 9 and Table 11, the tables indicate that the milling depth is highly significant in generating milling load and vibration acceleration, while the spindle speed is more significant in generating surface roughness.

## 4. Conclusions

In this study, the effects of process parameters such as milling depth, spindle speed and feed rate on the robotic milling performance were experimentally investigated during the flat-end milling of the 7075-T651 high-strength aeronautical aluminum alloy. The results of milling load, surface quality and vibration under each set of milling parameters were statistically compared. The contribution ratio based on the analysis of variance (ANOVA) was introduced to investigate the relation between parameters and milling performance. The conclusions of this paper are summarized as follows:

(1) Based on ANOVA and contribution ratio assessment, it has been estimated that milling depth (69.25%) has the highest percentage of influence on the milling load followed by feed rate (23.32%) and spindle speed (5.78%). The spindle speed has a high significance on surface roughness, with a contribution ratio of 48.02%, followed by milling depth (35.83%) and feed rate (13.18%).

(2) Milling depth is also highly significant in vibration. The contribution ratios of milling depth, spindle speed and feed rate on vibration in the X direction and the Y direction are basically equivalent, being 51.41%, 36.55% and 8.04% and 47.17%, 41.42% and 8.06%, while the contribution ratios on vibration in the Z direction are 75.97%, 13.43% and 8.16%, respectively.

(3) The significance and contribution ratio of the above process parameters on vibration acceleration is different in different directions. In detail, the contribution ratio of milling depth on axial vibration is quite different from the vibration in the feed direction and the vertical feed direction, being 51.41% in the X direction, 41.42% in the Y direction, and 75.97% in the Z direction. The contribution ratio of milling depth on vibration in the Z direction is much larger than the other two directions.

(4) From the analysis of the influence of feed per tooth on surface roughness, it has been found that the value of surface roughness decreases with increasing cutting speed for the same value of feed per tooth. The effect of cutting speed on the surface roughness is related to the formation of chips. However, surface roughness also depends from feed per tooth; it mainly increases with feed per tooth for stability process of cutting.

(5) Overall, the milling load, surface roughness and vibration acceleration all showed an increasing trend with the increase of milling depth and feed speed, and showed a decreasing trend with the increase of spindle speed.

## Figures and Tables

**Figure 1 materials-15-03566-f001:**
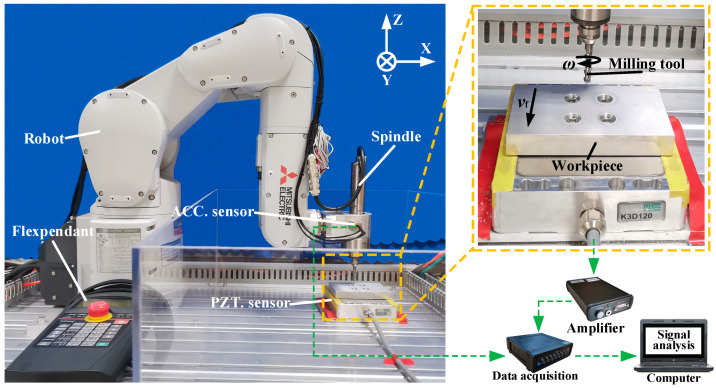
Schematic diagram of the experimental setup.

**Figure 2 materials-15-03566-f002:**
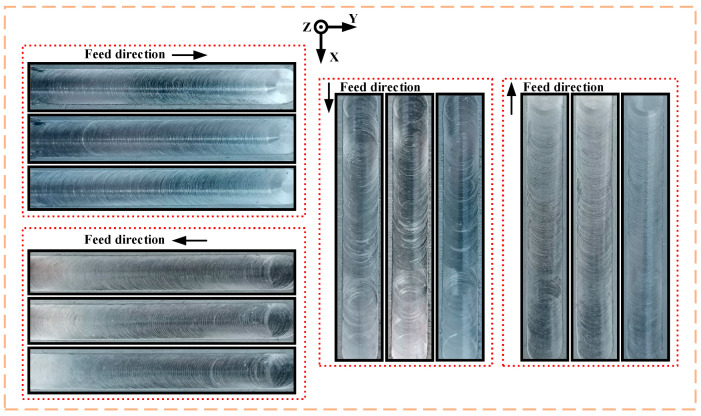
The machined surface results for milling by the robot in different directions.

**Figure 3 materials-15-03566-f003:**
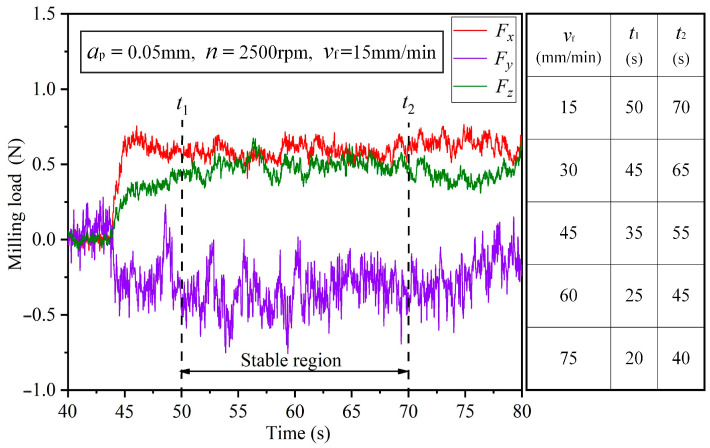
Milling load component in three directions.

**Figure 4 materials-15-03566-f004:**
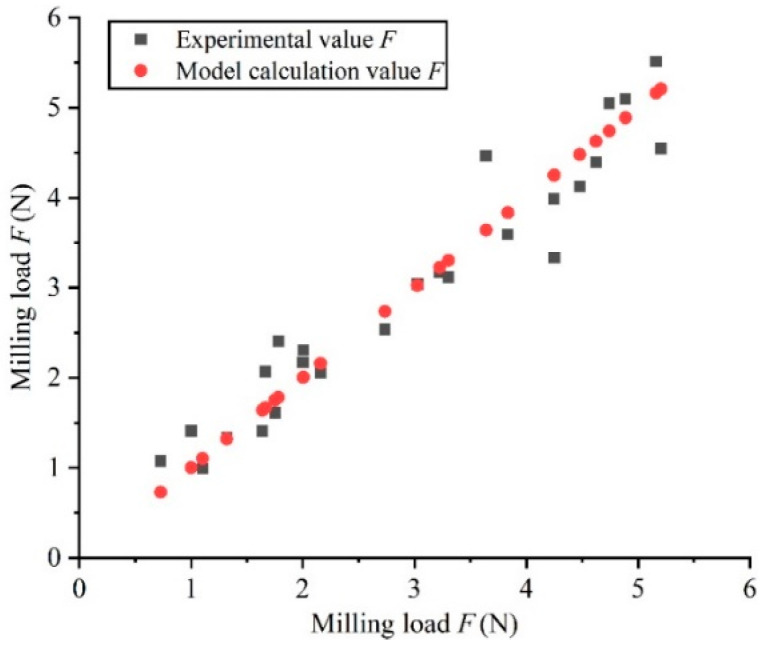
Comparison of model calculation value and experiment measurement value.

**Figure 5 materials-15-03566-f005:**
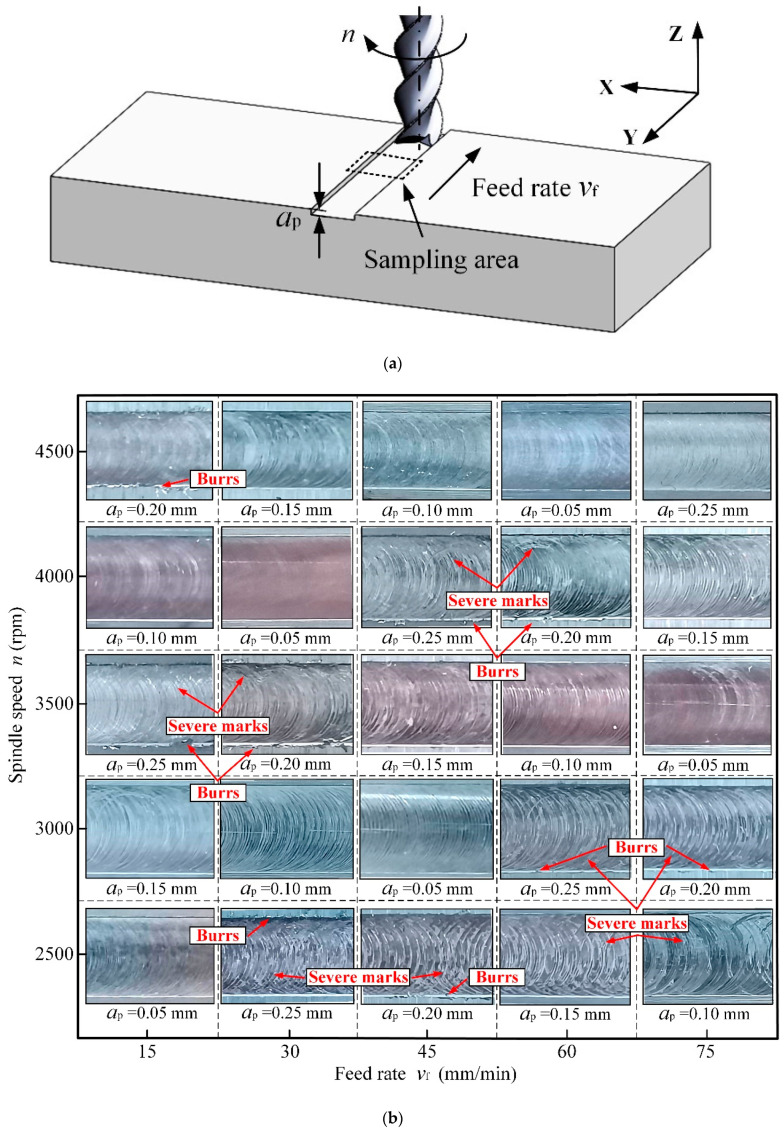
The machined surface morphology under each set of milling parameters. (**a**) Machined workpiece. (**b**) Summary of surface morphology in the sampling area.

**Figure 6 materials-15-03566-f006:**
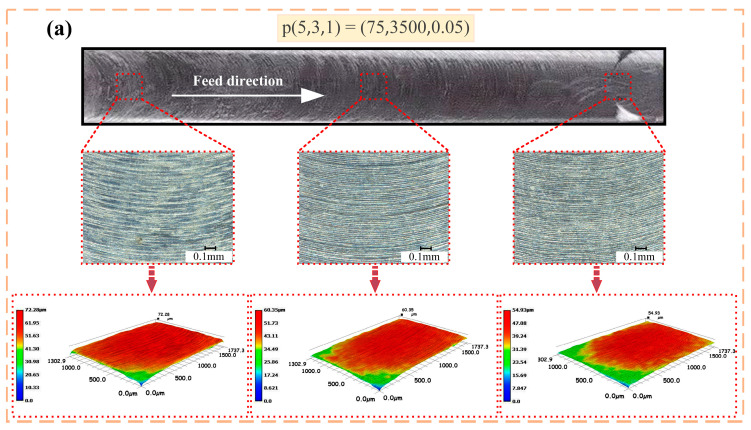
Photomicrographs of the machined surface morphology at different processing areas. (**a**) Experiment 3. (**b**) Experiment 8. (**c**) Experiment 13. (**d**) Experiment 18. (**e**) Experiment 23.

**Figure 7 materials-15-03566-f007:**
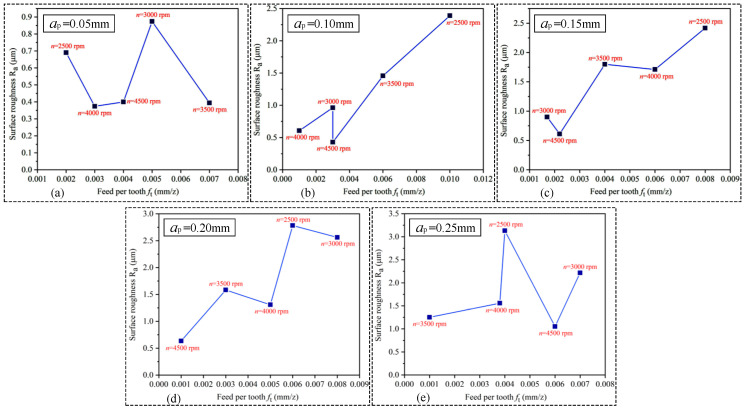
The effect of feed per tooth on surface roughness. (**a**) $a_{p}$ = 0.05 mm. (**b**) $a_{p}$ = 0.10 mm. (**c**) $a_{p}$ = 0.15 mm. (**d**) $a_{p}$ = 0.20 mm. (**e**) $a_{p}$ = 0.25 mm.

**Table 1 materials-15-03566-t001:** Chemical composition of 7075-T651.

Element	Ti	Si	Mn	Mg	Fe	Cr	Zn	Cu	Al
wt.%	0.2	0.4	0.3	2.1~2.9	0.5	0.18~0.28	5.1~6.1	1.2~2.0	Remainder

**Table 2 materials-15-03566-t002:** Material properties of 7075-T651.

Properties	Density (g/cm^3^)	Hardness (HB)	Yield Strength (MPa)	Tensile Strength (MPa)	Elastic Modulus (G Pa)	Elongation (%)
Value	2.81	150	503	572	71.7	11

**Table 3 materials-15-03566-t003:** Parameters of experimental system.

Parameters of Robot		Parameters of Spindle	
Properties	Value	Properties	Value
Body weight (kg)	41	Motor model	EM-3030T-J
Operating radius (mm)	649	Rated power (W)	350
Rated load (kg)	4	Rated speed (rpm)	7500
Repeatability (mm)	±0.02	Maximum speed (rpm)	30,000
The number of axis	6	Cooling method	Air cooling

**Table 4 materials-15-03566-t004:** Parameters of experiment and levels.

Factor	Level 1	Level 2	Level 3	Level 4	Level 5
$a_{p}$	0.05	0.10	0.15	0.20	0.25
n	2500	3000	3500	4000	4500
$v_{f}$	15	30	45	60	75

**Table 5 materials-15-03566-t005:** Milling experiment results of 7075-T651 aluminum alloy.

Test No.	Milling Depth (mm)	Spindle Speed (rpm)	Feed Rate (mm/min)	$\;F_{x}$ (N)	$F_{y}$ (N)	$F_{z}$ (N)	$F_{R}$ (N)	Surface Roughness $R_{a}$ (μm)	$a_{x}\;$ (g)	$\;\;a_{y}\;$ (g)	$a_{z}\;$ (g)
1	0.05	2500	15	0.593	−0.840	0.401	1.104	0.690	3.800	3.722	4.926
2	0.05	3000	45	0.974	−1.172	0.603	1.639	0.874	5.291	5.735	8.344
3	0.05	3500	75	1.040	−1.288	0.574	1.752	0.394	1.981	0.993	3.187
4	0.05	4000	30	0.469	−0.521	0.196	0.728	0.374	1.236	0.591	1.731
5	0.05	4500	60	0.774	−1.022	0.314	1.320	0.400	1.510	0.934	4.205
6	0.10	2500	75	1.957	−2.064	1.521	3.225	2.389	9.999	12.282	17.647
7	0.10	3000	30	0.993	−1.180	0.631	1.666	0.962	5.962	6.313	8.581
8	0.10	3500	60	1.312	−2.293	0.712	2.736	1.457	6.797	7.241	13.938
9	0.10	4000	15	0.462	−0.834	0.309	1.002	0.607	2.474	2.243	6.686
10	0.10	4500	45	1.273	−1.653	0.559	2.160	0.428	1.775	0.938	6.000
11	0.15	2500	60	2.616	−2.723	1.943	4.247	2.417	11.375	16.352	34.115
12	0.15	3000	15	0.875	−1.754	0.413	2.003	0.900	6.257	5.616	12.043
13	0.15	3500	45	1.754	−2.714	0.688	3.304	1.799	9.378	11.108	27.335
14	0.15	4000	75	2.179	−2.954	1.112	3.835	1.711	9.537	11.602	30.964
15	0.15	4500	30	0.978	−1.431	0.412	1.782	0.612	3.150	4.337	13.167
16	0.20	2500	45	2.360	−2.385	1.412	3.640	2.782	18.605	17.51	43.471
17	0.20	3000	75	3.088	−3.208	1.630	4.742	2.562	21.404	18.388	49.884
18	0.20	3500	30	1.689	−3.872	0.473	4.251	1.583	9.198	10.724	29.248
19	0.20	4000	60	2.347	−3.667	1.048	4.478	1.308	9.066	6.675	30.815
20	0.20	4500	15	0.946	−1.761	0.171	2.006	0.634	3.594	4.785	15.699
21	0.25	2500	30	2.516	−4.513	0.625	5.205	3.131	22.403	21.864	51.942
22	0.25	3000	60	3.488	−3.575	1.300	5.161	2.216	18.578	16.791	44.044
23	0.25	3500	15	0.610	−2.956	0.209	3.026	1.252	10.590	10.723	32.704
24	0.25	4000	45	2.382	−3.941	0.423	4.624	1.558	10.933	9.856	33.901
25	0.25	4500	75	3.102	−3.718	0.661	4.887	1.051	7.033	6.896	29.100

**Table 6 materials-15-03566-t006:** Variance analysis results of milling load orthogonal experiment.

Factor	SS	D	MS	F	Contribution (%)	Significance	Rank
$a_{p}$	33.728	4	8.432	41.968	69.25%	Highly significant	Milling depth-1 Feed rate-2 Spindle speed-3
n	2.814	4	0.704	3.501	5.78%	Not significant	
$v_{f}$	11.358	4	2.840	14.133	23.32%	Significant	
Error	2.411	12	0.201		1.65%		
Total	50.311	24					

**Table 7 materials-15-03566-t007:** Visual analysis results of surface roughness orthogonal experiment.

	Factor	Milling Depth	Spindle Speed	Feed Rate
Levels				
$k_{1}$		0.546	2.282	0.817
$k_{2}$		1.144	1.503	1.332
$k_{3}$		1.488	1.297	1.464
$k_{4}$		1.774	1.112	1.560
$k_{5}$		1.842	0.601	1.621

**Table 8 materials-15-03566-t008:** The feed per tooth in each experiment.

Test No.	Milling Depth (mm)	Spindle Speed (rpm)	Feed Rate (mm/min)	Feed Per Tooth (mm/z)	$\mathrm{Surface}\;\mathrm{Roughness}\;R_{a}$ (μm)
1	0.05	2500	15	0.002	0.690
2	0.05	3000	45	0.005	0.874
3	0.05	3500	75	0.007	0.394
4	0.05	4000	30	0.003	0.374
5	0.05	4500	60	0.004	0.400
6	0.10	2500	75	0.010	2.389
7	0.10	3000	30	0.003	0.962
8	0.10	3500	60	0.006	1.457
9	0.10	4000	15	0.001	0.607
10	0.10	4500	45	0.003	0.428
11	0.15	2500	60	0.008	2.417
12	0.15	3000	15	0.002	0.900
13	0.15	3500	45	0.004	1.799
14	0.15	4000	75	0.006	1.711
15	0.15	4500	30	0.002	0.612
16	0.20	2500	45	0.006	2.782
17	0.20	3000	75	0.008	2.562
18	0.20	3500	30	0.003	1.583
19	0.20	4000	60	0.005	1.308
20	0.20	4500	15	0.001	0.634
21	0.25	2500	30	0.004	3.131
22	0.25	3000	60	0.007	2.216
23	0.25	3500	15	0.001	1.252
24	0.25	4000	45	0.004	1.558
25	0.25	4500	75	0.006	1.051

**Table 9 materials-15-03566-t009:** Variance analysis results of surface roughness orthogonal experiment.

Factor	SS	D	MS	F	Contribution (%)	Significance	Rank
$a_{p}$	5.640	4	1.410	12.043	35.83%	Highly significant	Spindle speed-1 Milling depth-2 Feed rate-3
n	7.561	4	1.890	16.144	48.02%	Highly significant	
$v_{f}$	2.075	4	0.519	4.431	13.18%	Significant	
Error	1.405	12	0.117		2.97%		
Total	16.681	24					

**Table 10 materials-15-03566-t010:** Visual analysis results of vibration acceleration orthogonal experiment.

	Factor	Milling Depth	Spindle Speed	Feed Rate
Levels				
Vibration acceleration $a_{x}$				
$k_{1}$		2.764	13.236	5.343
$k_{2}$		5.401	11.498	8.390
$k_{3}$		7.939	7.589	9.196
$k_{4}$		12.373	6.649	9.465
$k_{5}$		13.907	3.380	9.991
Vibration acceleration $a_{y}$				
$k_{1}$		2.395	14.346	5.418
$k_{2}$		5.803	10.569	8.766
$k_{3}$		9.803	8.158	9.029
$k_{4}$		11.616	6.193	9.599
$k_{5}$		13.226	3.515	10.032
Vibration acceleration $a_{z}$				
$k_{1}$		4.479	30.420	14.412
$k_{2}$		10.570	24.579	20.934
$k_{3}$		23.525	21.282	23.810
$k_{4}$		33.823	20.819	25.423
$k_{5}$		38.338	13.613	26.156

**Table 11 materials-15-03566-t011:** Variance analysis results of vibration acceleration orthogonal experiment.

**Vibration acceleration** $a_{x}$							
Factor	SS	D	MS	F	Contribution (%)	Significance	Rank
$a_{p}$	435.312	4	108.828	12.884	51.41%	Highly significant	Milling depth-1 Spindle speed-2 Feed rate-3
n	309.439	4	77.360	9.158	36.55%	Highly significant	
$v_{f}$	68.076	4	17.019	2.015	8.04%	Not significant	
Error	101.362	12	8.447		4.00%		
Total	914.190	24					
**Vibration acceleration** $a_{y}$							
**Factor**	**SS**	**D**	**MS**	**F**	**Contribution (%)**	**Significance**	**Rank**
$a_{p}$	391.319	4	97.830	14.126	47.17%	Highly significant	Milling depth-1 Spindle speed-2 Feed rate-3
n	343.628	4	85.907	12.405	41.42%	Highly significant	
$v_{f}$	66.909	4	16.727	2.415	8.06%	Not significant	
Error	83.105	12	6.925		3.35%		
Total	884.961	24					
**Vibration acceleration** $a_{z}$							
**Factor**	**SS**	**D**	**MS**	**F**	**Contribution (%)**	**Significance**	**Rank**
$a_{p}$	4232.908	4	1058.227	31.128	75.97%	Highly significant	Milling depth-1 Spindle speed-2 Feed rate-3
n	748.435	4	187.109	5.504	13.43%	Significant	
$v_{f}$	454.423	4	113.606	3.342	8.16%	Not significant	
Error	407.952	12	33.996		2.44%		
Total	5843.718	24					

## Data Availability

Not applicable.

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
