# Peer review of "Contribution Ratio Assessment of Process Parameters on Robotic Milling Performance"

_materials, 2022, doi:10.3390/ma15103566_

Round 1

Reviewer 1 Report

The authors improved the paper properly, therefore, my decision about the paper is about to accept it.

Author Response

We are very grateful for your professional review work on this manuscript. Your suggestions for revision are very encouraging and helpful for us to improve the quality of our manuscript.

Once again, we sincerely thank you for your warm work. Your recognition of this paper is a great support for our research work.

Reviewer 2 Report

The scientific level of the paper is acceptable, but there are two major issues:

- the paper presents an experimental program meant to study the influence of cutting regime parameters upon milling load, surface quality and vibrations occurring during orthogonal milling. The use of robot as technological equipment has no relevance as long as only simple trajectories (straight lines) were generated. Moreover, the influence of the kinematics and dynamics of the robotic structure upon the process was not addressed in this study. Thus, the use of the term “robotic milling” is somehow forced, and the experimental studies are in fact linked only with a general milling process and its parameters.

- the conclusions of this study are general and can be obtained with virtually any material that can be machined through a milling process, even if the authors use a certain material for experiments. Thus, the paper does not bring any relevant contributions in the field of material science, rather in the filed of machining technology (and these are also straightforward, rather than novelties).

Author Response

Our response is given in normal font and the previous changes/additions to the revised manuscript are given in red text. The changes/additions to response the Reviewer #2 are given in purple text. 

Reviewer 3 Report

Article significantly improved but  I have some remarks to article. Please consider my comments to improve of article

Comment  1

In context of result of your research I propose also  add reference 

System to Surface Control in Robot Machining. AMR 2012;463–464:708–11. https://doi.org/10.4028/www.scientific.net/amr.463-464.708.

Comment 2

You describe Influence of spindle speed (cutting speed) on surface roughness

In explanation of results of surface roughness you wrote:

“Similar results are reported by literatures for the effect of process parameters on surface roughness [41]. The spindle speed (cutting speed) has found to be the most effective parameter on surface roughness. The effect of spindle speed on the surface roughness is related to the formation of chips. At low spindle speed, the formation of chip is slow. At high spindle speed, the chip is formed faster. This leads to a shorter time for the chips to be in contact with the newly formed surface of work- piece and the tendency for the chips to wrap back to the new face form is little as com- pared to low speed. In addition, the chip formation process is influenced by the shear length in the shear zone. And the shear length is related to the undeformed chip thickness and the shear angle [42]. The shear angle is large at high cutting speed. This leads to a smaller shear length. Therefore, the chip will break away with less plastic deformation, which in turn preserved the machined surface properties [43].”

Yes this is right but please also add  that

This is known effect of cutting speed on surface roughness due to an increase in cutting temperature. Please to add this explanation. Please also to add information that in your experiment cutting speed was changed in wide limits.

“The actual value of cutting speed ? ? corresponding to spindle speed is 47.1, 56.5, 66.0, 75.4 and 84.8 m / min"

The increase in speed results in a more reflective surface due to the disappearance of the built-up phenomenon (it also depends on the ap)

Comment 3

In conclusion you wrote

  1. Conclusions

The following conclusions were drawn from the present study:

(1) Based on ANOVA and contribution ratio assessment, it has been estimated that, milling depth (69.25%) has the highest percentage of influence on the milling load followed by feed rate (23.32%) and spindle speed (5.78%). And the spindle speed has a high significance on surface roughness….

I propose use of bullet:

  • Based on ANOVA and contribution ratio assessment, it has been estimated that, milling depth (69.25%) has the highest percentage of influence on the milling load followed by feed rate (23.32%) and spindle speed (5.78%). And the spindle speed has a high significance on surface roughness….

Author Response

Our response is given in normal font and the previous changes/additions to the revised manuscript are given in red text. The changes/additions to response the Reviewer #3 are given in blue text.

Round 2

Reviewer 2 Report

The authors have included some arguments backing-up the idea that the process studied through the experimental program presented in the paper could be considered "robotic milling", not "general milling". Thus, even if I do not entirely agree with those arguments, I consider that the title of the paper is now correct.
Moreover, some arguments pointed by the authors support the idea that the results of this study are valuable for materials science, thus one may consider that the paper ultimately fits into the thematic of journal.

This manuscript is a resubmission of an earlier submission. The following is a list of the peer review reports and author responses from that submission.

Round 1

Reviewer 1 Report

The authors presented the manuscript named “Contribution ratio assessment of process parameters on robotic 2 milling performance” which looks highly impressive in many aspects of machining. The paper can be accepted after the following items will be carried out:

  • Please demonstrate in the abstract novelty, practical significance.
  • How the authors calculate the vibration and cutting force components as they seem with a value in the table 5.
  • It seems that the authors used Taguchi method in the experiment table. Taguchi has some problems as it disappear some of the lines in the table. How the authors approach this situation?
  • Please don’t use abbreviations in the tables. The authors need to explain how cutting depth is effective on cutting forces rather than giving an example from the literature.
  • Literature studies show that feed rate is the most effective parameter on surface roughness. But there is a different result seen in here. The authors must explain this result with supportive result including the machining mechanism lies behind it.
  • Figure 5 need to be extended with different experiments.
  • There is a problem with the determination of the cutting parameters. Due to minor cutting depth is five times lower than the major cutting depth; it is highly possible that it is the biggest effect on machining characteristics. However, cutting speed was kept in a small range. This differences need to be focused while commenting the results.
  • Conclusions are short and insufficient. Please add numerical and key findings of the study.
  • There is a need to add more papers from MDPI journals. Some of them are recommended below:
  1. Performance evaluation of vegetable oil-based nano-cutting fluids in environmentally friendly machining of inconel-800 alloy
  2. Modeling of cutting parameters and tool geometry for multi-criteria optimization of surface roughness and vibration via response surface methodology in turning of
  3. Optimization and analysis of surface roughness, flank wear and 5 different sensorial data via tool condition monitoring system in turning of AISI 5140
  4. Investigations of machining characteristics in the upgraded MQL-assisted turning of pure titanium alloys using evolutionary algorithms
  5. The effects of MQL and dry environments on tool wear, cutting temperature, and power consumption during end milling of AISI 1040 steel
  6. Multi-objective optimization for grinding of AISI D2 steel with Al2O3 wheel under MQL

Reviewer 2 Report

This paper investigates about the contribution of the machining parameters such as milling depth, spindle speed, feed rate.

Based on the cutting force which was derived using experiments, various conditions are investigated in terms of surface quality and vibration.

Basically, milling machine should be investigated according to the various postures with respect to the compliance, which eventually results in surface roughness and vibration.

After all, this manuscript is short of scientific soundness.

Reviewer 3 Report

The experimental program presented in the paper is comprehensive, but I cannot see any contributions with regards of materials science in the paper. Consequently, it is questionable why the paper was submitted to Materials journal.

Some minor issues within the paper:

Line 34: rotating files – what does it mean?

Lines 206-207: the signal of the three milling load components was recorded by piezoelectric sensors and obtained the dynamic response diagram of the milling load over time – please provide a detailed explanation of how the data acquisition system was implemented and include a schematic diagram of it

Aside of these minor issues, there is a big question regarding the use of “robot milling” in the title. Even if a robot was used as technological equipment, the research was focused on the simple process of end milling, without really taking into consideration the use of the robotic structure.

All the experiments are unfolded for a flat-end milling process, on a simple planar surface. This application is not specific for robots, which due to their superior kinematic are intended for complex milling operations (sculptural surfaces).

The experimental program is a throughout one, but the conclusions are very general and valid for any end-milling process, no matter what technological equipment and material are used.

The relation with robotic milling is somehow forced, no discussion regarding how the kinematic and the dynamic of the robot structure are influenced by the cutting regime is presented. Moreover, the influence of the workpiece position in the workspace of the robot is not considered.

Reviewer 4 Report

Article present very interesting research of using industrial robots equipped in tool for subtractive manufacturing by milling. Paper present results of experiment prepared for three parameters on sily, surface roughness and vibrations

Paper need improvement for explanation of some detail of experiment. Also, an extension of the context using of industrial robot equipped with tool for manufacturing is desirable.

Please in review consider my remarks presented in comments below

Comment 1

Article present one of possibility to use of robot equipped in tool for machining. I propose to consider to add more references in scientific background of article  for present more possibilities  of robot applications in manufacturing, when robot is used as machine tool with large volume of machine space than conventional machine tools.   I propose to consider of add some references:

“Implementation of a robot system for sculptured surface cutting. Part 1. Rough Machining”. Int. J. Adv. Manuf. Technology, 1999, vol15, pp 624-629. doi:10.1007/s001700050111

“Manufacturing Using Robot.” Advanced Materials Research, 2012, vol. 463–464, pp. 1643–1646. Crossref, doi:10.4028/www.scientific.net/amr.463-464.1643.

“Printing Polymeric Materials for Robots with Embedded Systems”. Technologies 2021, 9, 82. https://doi.org/10.3390/technologies9040082

Comment 2

In paper you present parameters used in experiments: n  rot/min, f mm/min and ap mm 

n - spindle rotations this is technological parameter need for setting of spindle.  You ought to show also value  of cutting speed.

Please add also information of cutting speed. It is connected with n and tool diameter.In paper you wrote that you used mill with diameter of 6 mm. 

In my opinion you can  only describe that research was made for conditions when Vc was changed from vc= 47,1-84.8 m/min

Feed f mm / min does not represent the actual working conditions of the cutting edge, please provide also value on tooth Ft. It depends from n - spindle speed and number of teeth. In article you wrote that you use mill with  three teeth (blades) Based on this real parameters of cutting process you can analyze of obtain results of experiment  I propose also consider of real value of feed on tooth

Comment 3

I propose analyse of results based on real feed on tooth because it depends both from on feed and n you can get new explanation of results. Its known effect of  cutting speed on surface roughness  which decrease with cutting speed for the same value of feed on tooth. But surface roughness also depends from  feed on tooth mainly increase with feed of tooth for stability process of cutting.